# Ancestral Sperm Ecotypes Reveal Multiple Invasions of a Non-Native Fish in Northern Europe

**DOI:** 10.3390/cells10071743

**Published:** 2021-07-09

**Authors:** Leon Green, Apostolos Apostolou, Ellika Faust, Kajsa Palmqvist, Jane W. Behrens, Jonathan N. Havenhand, Erica H. Leder, Charlotta Kvarnemo

**Affiliations:** 1Department of Biological and Environmental Sciences, University of Gothenburg, Medicinaregatan 18A, 41390 Gothenburg, Sweden; kajsa.palmqvist@gmail.com (K.P.); lotta.kvarnemo@bioenv.gu.se (C.K.); 2Tjärnö Marine Laboratory, Linnaeus Centre for Marine Evolutionary Biology, University of Gothenburg, Laboratorievägen 10, 45296 Strömstad, Sweden; ellika.faust@marine.gu.se (E.F.); jon.havenhand@marine.gu.se (J.N.H.); erica.leder@marine.gu.se (E.H.L.); 3Gothenburg Global Biodiversity Centre, University of Gothenburg, Karl Skottbergs gata 22, 41390 Gothenburg, Sweden; 4Institute for Biodiversity and Ecosystem Research, Bulgarian Academy of Sciences, Tsar Osvoboditel Blvd. 1, 1000 Sofia, Bulgaria; apostolosfish@abv.bg; 5Tjärnö Marine Laboratory, Department of Marine Sciences, University of Gothenburg, Laboratorievägen 10, 45296 Strömstad, Sweden; 6National Institute of Aquatic Resources, Technical University of Denmark, Kemitorvet, Building 202, 2800 Kgs. Lyngby, Denmark; jabeh@aqua.dtu.dk; 7Natural History Museum, University of Oslo, Sars’ gate 1, 0562 Oslo, Norway

**Keywords:** gametes, Gobiidae, invasive, local adaptation, *Neogobius melanostomus*, NIS, Ponto-Caspian, round goby

## Abstract

For externally fertilising organisms in the aquatic environment, the abiotic fertilisation medium can be a strong selecting force. Among bony fishes, sperm are adapted to function in a narrow salinity range. A notable exception is the family Gobiidae, where several species reproduce across a wide salinity range. The family also contains several wide-spread invasive species. To better understand how these fishes tolerate such varying conditions, we measured sperm performance in relation to salinity from a freshwater and a brackish population within their ancestral Ponto-Caspian region of the round goby, *Neogobius melanostomus*. These two ancestral populations were then compared to nine additional invaded sites across northern Europe, both in terms of their sperm traits and by using genomic SNP markers. Our results show clear patterns of ancestral adaptations to freshwater and brackish salinities in their sperm performance. Population genomic analyses show that the ancestral ecotypes have generally established themselves in environments that fit their sperm adaptations. Sites close to ports with intense shipping show that both outbreeding and admixture can affect the sperm performance of a population in a given salinity. Rapid adaptation to local conditions is also supported at some sites. Historical and contemporary evolution in the traits of the round goby sperm cells is tightly linked to the population and seascape genomics as well as biogeographic processes in these invasive fishes. Since the risk of a population establishing in an area is related to the genotype by environment match, port connectivity and the ancestry of the round goby population can likely be useful for predicting the species spread.

## 1. Introduction

Since the onset of Parker’s development of sperm evolutionary theories in the 1970’s (most notably Parker 1970 [1]), the understanding of and research on gamete evolution has increased dramatically [2,3,4]. Though much of the work has focused on the evolution of the sexes [5,6], their gametes’ form and function [7,8,9], and the importance of sexual selection on sperm evolution [10,11,12], an understanding of the role of gametes during contemporary evolution is still in its infancy. A few recent studies have shown that sperm traits can rapidly adapt to biotic [13] and abiotic conditions [14,15], and even shape speciation events [16,17]. Rapid adaptation of spermatozoa makes sense in light of the sensitivity of gametic life stages to the ambient environment [18]. For organisms whose fertilization events take place outside their bodies, the lack of a protective reproductive tract with a comparatively benign environment means sperm are even further exposed to external stressors [19,20,21].

For externally fertilizing fish (~98% of teleost species, [19]) sperm movement and velocity is generally triggered by the osmolality, or in some species, ions associated with a specific osmolality, of the surrounding water [22]. Since environments vary in their osmolality by the salt content of the water, fish sperm are often adapted to function in a specific salinity or within a very narrow range of salinities [23]. A few species that can reproduce across a wide range of salinities are able to respond plastically to the environment and produce sperm that are triggered to swim in the salinity in which the males have been spending the last days or months (depending on species) [24,25,26]. Interestingly, several of these fishes have the status of non-indigenous species (NIS), such as the killifish [26] and *Tilapia* spp. [25].

The spread of NIS is increasing as a consequence of increased transportation and trade across the globe. This is true not only for species that are intentionally transported, for example as part of pet or plant trade, but perhaps more importantly for the myriad of NIS that are transported unintentionally, stowed away among farm produce [27], in packaging materials [28], or even as zoonotic passengers inside other animals [29]. Aquatic NIS often follow ballast water discharge into novel habitats and are dependent on a phenotypic match with the new environment in order to survive and reproduce. The new environment may also affect the introduced organism differently during different life-stages. Consequently, the match needs to cover the entire phenotype, from zygote to juvenile to adult, including the gametic life stages of reproducing adults and their offspring.

One of the most successful NIS is the round goby (*Neogobius melanostomus*, Pallas 1811 [30]). Originating from the Ponto-Caspian region, it is now invasive on both sides of the Atlantic Ocean. This perciform fish has spread rapidly outside its ancestral range since the early 1990s with varying ecological impacts reported in the wake of its invasion [31,32]. The species now covers a wide range of salinities from freshwater to fully marine (≥25 practical salinity units, PSU) [33,34]. Theory predicts that selection on sperm in the round goby is strong, since they often reproduce under sperm competition, and detrimental environmental conditions may cause sperm limitation [35,36,37]. Recent studies suggest that the round goby is incapable of acclimating their sperm performance to match novel salinity conditions [38], but there is support for local adaptation in its sperm traits [39]. This has raised the hypothesis that reproduction in widely different environments may be possible due to different ancestral origins of the populations that are now spreading in different aquatic environments within Europe and North America. This could result from a combination of phenotypic sorting and ongoing selection at large geographic scales, which in turn can be fueled by established trait adaptations of the ancestral ecotypes. Support for this can be found in the few existing studies of the species population genetics, which shows that round gobies in the freshwater Great Lakes in North America have their origin in Ponto-Caspian freshwater rivers [40], whereas populations in the brackish Baltic Sea originate from the brackish Black Sea [41]. The round goby in the Great Lakes and the Baltic Sea also seem to be well-adapted in terms of their sperm traits [39,42].

In order to better understand the link between sperm traits and the ancestry of different round goby populations, and how this may relate to potential establishment and spread in novel environments, we recorded sperm traits and sampled DNA from round gobies collected at a freshwater and a brackish site within their ancestral Ponto-Caspian region. We then compared our results to previously published sperm data and novel genotype data from nine introduced sites with different salinities (Table 1). Our aim was to test the predictions: (1) that sperm performance is locally adapted to salinity in fish from different salinity environments within the ancestral region; (2) that genotypes from both the freshwater and brackish source populations exist within the species invaded range; and (3) that the sperm of these genotypes show similar phenotypic responses to salinity as the ancestral populations.

## 2. Materials and Methods

### 2.1. Sampling Sites in the Ancestral Range

The ancestral region of the round goby (*Neogobius melanostomus melanostomus,* hereafter *N. melanostomus*) is the Black Sea and its tributaries [43]. In this study, round gobies were sampled during their breeding period in April–May 2017 from two sites in this region, the southern coast of Bulgaria (brackish) and the Danube river (freshwater) (Figure 1, Table 1). These sites were chosen because a previous population genetic study has shown that at least the first observed introduction into the Baltic Sea in 1990 in the Bay of Gdansk [44] is from the Bulgarian coast in the Black Sea [41]. The same study also distinguished round gobies sampled in the river environments of the Dnieper and the Danube from the populations in the Black Sea. The lower Danube is closely situated to western European waterways, and introduced fish in the upper Danube share close ancestry with sampled freshwater rivers from the native range [41]. The Danube is therefore a likely source for populations now inhabiting rivers in western Europe.

### 2.2. Sampling of Sperm and Reproductive Tissues from the Ancestral Range

Sperm measurements were performed following the same protocol as used in the previous studies to which they were to be compared against ([35,38,39]; Table 1). Fourteen males caught in Danube (freshwater origin, 0 PSU: N = 14) and twelve males caught near Sozopol on the Bulgarian coast (brackish origin, ~16 PSU: N = 12) were kept in aerated tanks with water from their site of capture for less than 7 days, before they were sampled for sperm data at the Institute of Biodiversity and Ecosystem Research, Bulgarian Academy of Sciences’ Laboratory of Marine Ecology, Sozopol, Bulgaria. Each male was checked for reproductive readiness by visual inspection of an erect genital papilla (following [42]), and visually scored to a reproductive tactic category (nest-holder; sneaker or intermediate, based on skin pigmentation and morphology [35]) and euthanized by a concussive blow to the head, followed by destruction of the brain. After this, the testes were dissected from the fish within 1 min, using stainless steel forceps and scissors (curved, sharp point, 4 inches, Sigma-Aldrich Co, St. Louis, MO, USA), weighed on a digital scale and one testis was placed into a 1.5 mL microcentrifuge tube (Eppendorf, Hamburg, Germany). The testis was incised five times using scissors to release sperm and the content was diluted with 60 μL calcium-free Ringer’s solution at 18 °C [45] for a roughly double increase in liquid volume and to prevent sperm activation. After this, the sample was stirred with a vortex (Vortex-Genie 2, Scientific Industries, Bohemia, NY, USA) for three consecutive stirs of one second each in rapid succession and then kept in a water bath at 18 °C until used within 10 min.

Sperm from this suspension were activated in five test salinities (0, 8, 16, 24 and 32 PSU), which were achieved by mixing river water taken from the Danube (site of collection: 44°03′13.7″ N 26°37′01.6″ E), distilled water and saline water taken from the brackish Black Sea (site of collection: 42°25′28.3″ N 27°41′57.9″ E, salinity at time of collection: 17.6 PSU). The river water was first heated to 100 °C in a glass vial, left to cool to room temperature and then vacuum filtered through a 0.7 µm filter (Whatman glass microfiber filters, GF/F grade). To create saline water for treatments, the water from the Black Sea was first heated to 100 °C in a glass vial and kept heated until the volume had reduced to 55% and 32 PSU salinity (which occurred in about 30 min), whereafter the water was left to cool, controlled for salinity and then filtered (as above). The five test salinities were as follows: 0 PSU (100% river water); 8 PSU (50% river water + 25% saline water + 25% distilled water); 16 PSU (50% saline water + 50% distilled water); 24 PSU (75% saline water + 25% distilled water); 32 PSU (100% volume reduced saline water). Water was kept frozen in 200 mL glass jars between use to eliminate the risk of microbial growth and thawed overnight before use.

Sperm were activated in each test salinity in randomized order and immediately prepared for filming in the following way: 25 µL of the sperm sample was transferred to a tube of 750 µL of one of the five different salinities. This diluted sperm suspension was then stirred with a vortex (as described above) and 45 μL of the suspension was transferred to a 2% (*w*/*v*) albumin coated glass slide fitted with an O-ring and covered with an albumin coated coverslip to form a suspended drop [46]. This was repeated for six technical replicates per male and test salinity. A video was recorded of each drop using a high-speed video camera (PixeLINK PL-D725, Ottawa, ON, Canada) fitted to an inverted microscope (Axio Vert.A1, Carl Zeiss AG, Oberkochen, Germany). Focus of the camera was set at the middle of the suspended drop, and sperm movement was filmed for 15 frames using a 10× magnification objective and standard contrast and illumination (30 frames s^−1^, size 2592 × 2048 pixels, exposure time 10 ms, gain 0, gamma 0.1). Sperm movement parameters from a total of 1,798,933 sperm tracks in a total of 1015 videos were extracted using the CASA plugin [47] for ImageJ (National Institutes of Health, Bethesda (Maryland), United States) following a protocol outlined in Green, Havenhand et al. (2020) [39]. Sperm were only classified as swimming if velocity was higher than 25 μm s^−1^, to avoid including non-swimming particles drifting in micro-currents.

### 2.3. Compiling Sperm Data from the Introduced Range

Data on sperm motility and sperm velocity tested in freshwater (0 or 1 PSU), brackish (15 or 16 PSU) and marine (30 or 32 PSU) salinities were collected for a total of nine sites from three previously published sources where raw data were available (Green, Havenhand, et al., 2020 [39], Green, Niemax, et al., 2020 [35], 2021 [38]). See Table 1 for details on each site and test salinities.

### 2.4. DNA Sampling and GBS Library Preparation

Fin samples were taken within 5 min of euthanasia of the fish from all sampled sites (Table 1) at the time of sperm sampling using steel scissors and forceps. The samples were immediately put into individually marked 1.5 mL Eppendorf tubes containing 99.7% ethanol and stored at −15 °C until extractions took place. DNA was extracted from the fin clips using the DNeasy extraction kit (Blood & Tissue Kit, Qiagen, Hilden, Germany) following the manufacturer’s protocol. Briefly, tissue (fin samples) of between 100 and 250 µg were lysed in Buffer ATL and Proteinase K at 56 °C for 4 h. Buffer AL was added to the lysate, and the lysate was vortexed and then pipetted into a spin column where it was spun down at 6000 g for 1 min. Buffer AW1 was then added and the column was washed at 6000 g for 1 min; Buffer AW2 was then added for a second wash at 20,000 g for 3 min. The DNA was then eluted with Buffer AE at 6000 g for 1 min. The concentration of the eluted sample was measured with a fluorometer (Qubit 3.0, Thermo Fisher Scientific, Waltham (Massachusetts), United States) and the sample diluted with lab-grade distilled water to decrease sample concentration to 100 ng in a volume of 16.5 µL.

Five separate genotype libraries were prepared and sequenced on two occasions, in 2017 and 2019 (Table 1). Library preparation followed a genotyping by sequencing (GBS) protocol [48] previously used successfully for related species [49]. In short, 1.5 µL of restriction enzyme (Pst1 HF 20 U/mL, New England Biolabs, Ipswich (Massachusetts), United States) and 2 µL of Cut Smart Buffer were added to 100 ng of DNA for each individual sample for digestion in a 20 µL reaction. The sample was then digested in a thermocycler using the following program: 1 h at 37 °C; 15 min at 75 °C; 10 min at 4 °C.

A unique forward adapter and a common reverse adapter were added to the digested DNA (1 µL of 50 nM pooled forward and reverse adapters), along with 21 µL dH_2_O, 5 µL T4-buffer, and 3 µL T4-ligase (New England Biolabs, Ipswich (Massachusetts), United States). Ligation was performed for 1 h at 22 °C, then stopped by incubation at 30 min at 65 °C and brought to 4 °C.

The ligation products from 96 individual samples with unique barcodes were then pooled into libraries (15 µL from each individual). Individuals from the same population were distributed across multiple libraries to avoid potential biases (see Table 1 for allocation per library). Libraries were cleaned and concentrated using Ampure XP beads and PCR amplified with the following components: cleaned ligation pool (23.5 µL), 1.5 µL primer mix (12.5 µM) and 25 µL KAPA HIFI Hot start 2X Mastermix. This was accomplished with the following amplification program: 5 min at 72 °C; four cycles of [30 s at 95 °C; 10 s at 95 °C; 30 s at 65 °C; 30 s at 70 °C]; 13 cycles of [10 s at 95 °C; 30 s at 65 °C; 20 s at 72 °C]; 5 min at 72 °C; 10 min 4 °C). Following the PCR amplification, each library was cleaned using Ampure XP (Beckman Coulter Inc., Brea, CA, USA) and then selected for a size range of 290–390 bases (run in duplicate) using the BluePippin system (Sage Biosciences, Beverly, MA, USA) with 2% gel cassette and V1 marker. Final cleaning and pooling of the two BluePippin elutions per library was performed using Ampure XP beads, with elution in 10 mM Tris-HCl. The final size was then confirmed on a Fragment Analyzer using the High Sensitivity Genomic DNA kit (DNF-488) (both from Agilent, Santa Clara, CA, USA). Final libraries had a mean fragment size of between 350 and 380 bases.

### 2.5. Sequencing and SNP Calling

The 5 GBS libraries were sequenced at 100 paired-ends using the Illumina HiSeq 4000 platform at the Beijing Genome Institute.

Cutadapt v2.10 [50] was used to demultiplex and remove barcodes and then remove low-quality bases and leftover adapter sequences. If read pairs had one read with low-quality ends, only read pairs with reads greater than 50 bases were kept in the analysis. Paired-end reads for each individual were aligned to the round goby reference genome (Ensembl release 100) using Bowtie2 v2.3.5.1 [51]. Sequence alignment map (SAM) files were subsequently converted to sorted binary version (BAM) files using SAMtools v1.10 [52], and variants were called and filtered using BCFtools mpileup, call, and filter functions (v1.10.2) [53].

Sites at which BCFtools identified multiple variant types (single nucleotide polymorphisms (SNPs), indels and multi-base polymorphisms) were removed. Individuals with >25% loci with no reads were removed from the dataset. All SNPs were then filtered for read depth of 6 and a minimum genotype quality of 30 and set to 0 (./.) if this was not met. Final filters were applied using BCFtools and Plink v1.07 [54], removing any SNPs with hwe < 0.001 (run separately on each site), missing genotypes > 0.1, minor allele frequency < 0.01 or heterozygosity > 0.75. The final dataset analysed in this study consisted of 13,847 SNPs across 264 individuals. The total numbers of individuals per site after filtering are presented in Table 1.

### 2.6. Analysis of Sperm Traits and Reproductive Tissues from the Ancestral Range

Analysis of sperm traits of fish from the ancestral range was performed on sperm motility (proportion motile sperm) and sperm velocity (velocity of the curvilinear path, VCL, μm s^−1^). Measurements of these traits allowed for comparison with the previously published data on round goby sperm traits of fish from introduced areas that are included in the current study (Table 1). Since each male’s sperm tested in a given salinity were filmed in six technical replicates, an average value was calculated from these and this value was used for the analysis. All statistical modeling was conducted in R (version 3.6.2) using the packages lme4 [55], lmerTest [56] and car [57]. Main predictor variables modelled as fixed effects were parent origin (freshwater or brackish) and sperm test salinity (0; 8; 16; 24; or 32 PSU). These were modelled in a full factorial design with all interactions available. The linear models were visually explored by inspecting the residuals vs. fitted values, theoretical and observed quantiles, high influence points, and the frequency distribution of residuals, all using the ‘plot(model)’ function. The models were also assessed for variance inflation factors using the ‘vif(model)’ function, and none of concern were found. Both response variables (sperm motility and sperm velocity) were normally distributed and met assumptions of sphericity. Since sperm from each male were tested in multiple salinities, male identity was included as a random factor to control for the repeated measurements. The influence of the random factor was tested by a comparison between the linear and mixed models using the ‘anova(linearmodel, mixedmodel)’ function. For both response variables the differences were significant and the random term was kept in each respective model. Mixed models were consistently found to have better fit than the linear ones and the random factor was therefore kept in the final models. Where *p* values were of interest, lmer models were fitted with an estimated *p*-value using Satterthwaite approximation in the R package lmerTest. Similarly, differences between environments in reproductive tissues were analysed with linear models following the above methods. A calculated gonadosomatic index (GSI: testes weight/body weight ∗ 100) was tested as response variable, and reproductive tactic (nest-holder, sneaker or intermediate) and origin (freshwater or brackish) as predictor variables.

### 2.7. Analysis of Sperm Traits between the Ancestral and Introduced Ranges

Sperm traits across sites sampled in previous studies, and the ancestral sites sampled for this study, were compared using linear models. Since test salinities between studies differed slightly, velocity and motility responses were grouped according to four salinity categories as follows: freshwater: sperm tested in 0–1 PSU; brackish: sperm tested in 15–16 PSU; marine: sperm tested in 30–32 PSU; local: the test salinity closest to the salinity at the males’ site of capture. Differences between groups were tested using linear models for each salinity category, with site as predictor variable and sperm velocity or sperm motility as response variable using the lm function in R. For the freshwater and brackish categories, the sites of the Danube (ancestral freshwater) and Sozopol (ancestral brackish) were chosen as ‘controls’, respectively, with the a priori expectation that fish from these sites were most likely to be locally adapted. For the marine and local categories, differences were simply tested between groups.

### 2.8. Genotype Analyses

Genetic diversity and divergence indices were calculated using the R package diveRsity [58]. Deviations from expected heterozygosity (He) were assessed by calculating F_IS_ according to Weir & Cockerham [59]. Deviations from Hardy–Weinberg (HW) proportions were estimated with exact tests, with *p*-values calculated according to the complete enumeration method [60]. Weir & Cockerham’s F_ST_ was estimated for each population pair. Statistical significance of F_ST_ values was assessed using Fisher’s exact probability test with 5000 Monte Carlo replicates, followed by false discovery rate correction [61]. Finally, pairwise F_ST_ estimates were visualized using a neighbor-joining tree.

To estimate and visualise genetic differentiation among individuals we applied two individual-based clustering methods using the ‘snmf’ function in the R Package LEA [62] and principal component analysis (PCA) in the R package ade4 [63,64]. The snmf function estimates individual ancestry by utilizing a sparse non-negative matrix factorization algorithm (sNMF) to compute least-squares estimates of ancestry coefficients. The ancestry coefficients represent the proportions of each individual’s genome that originated from a specified number of ancestral populations K. The function also estimates an entropy criterion which can be used to evaluate the quality of fit of the statistical model to the data using a cross-validation. To evaluate the number of ancestral populations that best explains the genotypic data ancestry, coefficients were estimated by running 6 replicates of K 1–20 using the cross-entropy criterion (CEC). The best-supported ancestry coefficient was selected based on the lowest CEC. PCA uses a multivariate exploratory approach that makes no prior assumptions about how many populations exist or boundaries between them. Allele frequencies were centered but not scaled and missing data were replaced by mean allele frequencies with the function scaleGen in adegenet [65,66].

Finally, to correlate F_ST_ and salinity, a Mantel test of the F_ST_ and salinity difference matrices, using 9999 permutations, was computed with the package ade4 in R [63,64].

### 2.9. Shipping Intensity

Shipping data (highest value of total number of vessels passing within 25 km of the site of catch, and distance to this point in km) for the Baltic sites (7 of the 9 introduced sites) were extracted from Helcom (http://maps.helcom.fi/website/aisexplorer/, accessed on the 10 June 2021) for the years 2006–2016. Data on shipping intensity in Elbe was approximated from statistics presented for 2013, 2017, 2018 and 2019 from the Port of Hamburg, Germany (https://www.hafen-hamburg.de/en/statistics/calls/, accessed on 14 June 2021). Data from Rhine (closest ports of Arnhem and Nijmegen, The Netherlands) were not publicly available upon contact and the site was therefore omitted from the analysis. The correlation between shipping intensity as well as distance to the point of highest shipping density within 25 km was then tested for correlations with population genetic values (results Section 3.4) using Pearson’s correlation tests in R.

## 3. Results

### 3.1. Sperm Performance in the Ancestral Region

In line with our predictions, sperm responses to test salinities were significantly different between male round gobies of freshwater and brackish origin from the ancestral range. More specifically, sperm velocity in the five test salinities depended on the origin of the fish (lmer, sperm test salinity x origin; F_(1, 24.7)_ = 45.89, *p* < 0.001) (Figure 2a). Sperm from fish of freshwater origin had the highest velocity in 0 PSU (freshwater), while sperm from fish of brackish origin had the highest velocity in 16 PSU (i.e., their local salinity).

Similarly, sperm motility in different test salinities was also found to be dependent on origin (lmer, sperm test salinity x origin; F_(4, 35.76)_ = 35.76, *p* < 0.001) (Figure 2b). Again, fish of freshwater origin had the highest percentage of motile sperm in 0 PSU (freshwater), while sperm from fish of brackish origin were the most motile in 16 PSU (i.e., their local salinity).

### 3.2. Comparing Sperm Performance between Ancestral and Introduced Sites

When comparing sperm performance (velocity and motility) between all ancestral and introduced sites within four categorical conditions (freshwater, brackish, marine and local), we found differences in sperm velocity between sites in all four conditions (freshwater, lm, F_(10,86)_ = 4.05, *p* < 0.001, Adj. R^2^ = 0.241; brackish, lm, F_(10,87)_ = 15.46, *p* < 0.001, marine; lm, F_(9,65)_ = 5.27, *p* < 0.001, Adj. R^2^ = 0.422; local; lm, F_(10,87)_ = 18.24, *p* < 0.001, Adj. R^2^ = 0.677).

Differences in sperm motility were found in three of four comparisons: freshwater (lm, F_(10,86)_ = 5.31, *p* < 0.001, Adj. R^2^ = 0.382); brackish (lm, F_(10,87)_ = 22.25, *p* < 0.001, Adj. R^2^ = 0.719); and local (lm, F_(10,87)_ = 16.95, *p* < 0.001, Adj. R^2^ = 0.661), but not in marine salinities (30–32 PSU), where site did not have a significant effect on motility (lm, F_(9,65)_ = 2.01, *p* = 0.052, Adj. R^2^ = 0.110). Significances for comparisons between sites are annotated in Figure 3. Detailed statistics can be found in Appendix A.

#### 3.2.1. Sperm in Freshwater Conditions (0–1 PSU)

Sperm from all sites were tested in freshwater conditions of 0–1 PSU, and this data was compared against the ancestral freshwater site of the river Danube (mean velocity: 62.89 µm^−s^, CI: 57.50–68.29; mean motility: 0.095, CI: 0.04–0.15; Figure 3a,b, Panel 1). The highest sperm velocity was found in males from this site. Sperm velocity of males from the river Rhine (freshwater) and the northern Baltic Sea sites of Raahe and Turku (low salinity brackish water) did not differ significantly from that of the Danube. Fish from all other sites had significantly lower sperm velocity in freshwater, including fish from the freshwater site of the river Elbe.

The proportion of motile sperm (sperm motility) was significantly higher in two introduced sites: the northern Baltic site of Turku (est. ± SE: 0.21 ± 0.05, *t* = 4.23, *p* < 0.001) and southern Baltic Karrebaeksminde (est. ± SE: 0.10 ± 0.05, *t* = 2.15, *p* = 0.034) compared to the ancestral freshwater site of the Danube. Sperm motility was significantly lower in southern Baltic Guldborgsund (est. ± SE: −0.09 ± 0.04, *t* = −2.18, *p* = 0.031) and in the ancestral brackish Black Sea site of Sozopol (est. ± SE: −0.08 ± 0.04, *t* = −2.11, *p* = 0.038) as compared to the ancestral river Danube.

#### 3.2.2. Sperm in Brackish Conditions (15–16 PSU)

In the brackish test conditions of 15–16 PSU, sperm were compared against the ancestral Black Sea site of Sozopol (mean velocity: 102.87 µm^−s^, CI: 91.14–114.60; mean motility: 0.125, CI: 0.08–0.17; Figure 3a,b, Panel 2). Overall, there were few differences in sperm velocity among the Baltic Sea sites and the ancestral site, with only the southern Baltic site of Travemünde showing a significantly higher velocity (est. ± SE: 18.42 ± 7.47, *t* = 2.47, *p* = 0.016). Fish from the two introduced freshwater sites showed significantly lower sperm velocity than fish from the Black Sea (Elbe: est. ± SE: −30.81 ± 11.80, *t* = −2.61, *p* = 0.011; Rhine: est. ± SE: −25.00 ± 10.88, *t* = −2.98, *p* = 0.024) and fish from the ancestral freshwater Danube had the lowest sperm velocity among all sites (est. ± SE: −61.47 ± 8.04, *t* = −7.64, *p* < 0.001).

When comparing sperm motility, fish from sites in the southwestern Baltic Sea i.e., Kindvig (est. ± SE: 0.41 ± 0.04, *t* = 9.78, *p* < 0.001) and Karrebaeksminde (est. ± SE: 0.21 ± 0.04, *t* = 4.92, *p* < 0.001) showed higher motility than fish from the ancestral brackish Sozopol. Significantly lower sperm motility was found for fish from the freshwater sites of Rhine (est. ± SE: −0.10 ± 0.04, *t* = −2.12, *p* = 0.037) and ancestral Danube (est. ± SE: −0.12 ± 0.03, *t* = 3.59, *p* < 0.001).

#### 3.2.3. Sperm in Marine Conditions (30–32 PSU)

In marine test conditions of 30–32 PSU, sperm performance was compared against the global mean (mean velocity: 58.86 µm^−s^, CI: 54.49–63.22; mean motility: 0.040, CI: 0.03–0.06; Figure 3a,b, Panel 3) across all sites except Travemünde (which lacked data for sperm tested in these conditions). The only site that showed a difference in sperm velocity was the ancestral freshwater Danube where velocity was significantly lower than the mean (est. ± SE: −27.64 ± 6.20, *t* = −4.46, *p* < 0.001).

The only site that showed a difference in sperm motility was Kindvig in the Danish Straits where motility was significantly higher than the mean (est. ± SE: 0.09 ± 0.03, *t* = 2.71, *p* = 0.009).

#### 3.2.4. Sperm Performance at Each Site’s Local Salinity

In local test conditions, sperm performance was measured in the test salinity most closely resembling the local conditions at each site (reported in each specific study and Table 1) and compared against the global mean across all sites (velocity: 94.23 µm^−s^, CI: 88.41–100.07; motility: 0.207, CI: 0.17–0.23; Figure 3a,b, Panel 4). These values are presented in the brackets following each site’s name on the x-axis in Figure 3.

Among brackish sites, fish from Travemünde in the southern end of the Baltic Sea showed higher sperm velocity (est. ± SE: 18.42 ± 6.37, *t* = 2.89, *p* = 0.005), while two sites from the northern Baltic Sea showed lower velocity compared to the mean: Mariehamn (est. ± SE: −23.22 ± 7.47, *t* = −3.11, *p* = 0.003), and Raahe (est. ± SE: −24.39 ± 8.30, *t* = −2.94, *p* = 0.004). All three freshwater sites showed significantly lower sperm velocity than the mean: the Elbe (est. ± SE: −52.65 ± 10.07, *t* = −5.23, *p* < 0.001), the Rhine (est. ± SE: −46.92 ± 9.29, *t* = −5.05, *p* < 0.001), and the ancestral Danube (est. ± SE: −39.97 ± 6.86, *t* = −5.82, *p* < 0.001).

Sperm motility did not show the same patterns, with four introduced brackish sites showing increased motility compared to the mean: Kindvig (est. ± SE: 0.41 ± 0.05, *t* = 8.16, *p* < 0.001), Karrebaeksminde (est. ± SE: 0.21 ± 0.05, *t* = 4.11, *p* < 0.001), Mariehamn (est. ± SE: 0.24 ± 0.04, *t* = 5.57, *p* < 0.001), and Turku (est. ± SE: 0.28 ± 0.05, *t* = 5.13, *p* < 0.001). The ancestral brackish site of Sozopol showed slightly lower sperm motility in the comparison (est. ± SE: −0.08 ± 0.03, *t* = −2.78, *p* = 0.007).

### 3.3. Differences in Reproductive Tactics in the Ancestral Region

Among the males caught from the ancestral freshwater site of the river Danube, no males were classed as sneakers based on morphological scoring. However, three males were classed as “intermediate” (based on a lack of dark colouration and “puffy” cheeks typical for nest-holder males, and also a lack of morphology suggesting sneaker tactics) [42]. Among the twelve males caught from the ancestral brackish site of Sozopol, six were classed as potential sneakers, and six were classed as nest-holders. Analysis of GSI showed that among the males of the two classes caught in Danube, “intermediate” males did not show significantly different GSI compared to the males classed as nest-holding (est. ± SE: −0.50 ± 0.35, *t* = −1.45, *p* = 0.17). In Sozopol, however, males classed as sneakers had significantly higher GSI (GSI: 4.59, CI: 1.56–4.08) than males classed as nest-holders (est. ± SE: 2.82 ± 0.57, *t* = 4.98, *p* < 0.001). There was no difference in GSI between males classed as nest-holders from the freshwater (GSI: 1.61, CI: 1.26–1.96) and brackish (GSI: 1.77, CI: 0.88–2.66) ancestral sites (est. ± SE: 0.16 ± 0.25, *t* = 0.67, *p* = 0.52).

### 3.4. Population Genetic Structures between Ancestral and Introduced Sites

The overall genetic diversity was highest in the Sozopol sample, which had the highest number of polymorphic sites and private alleles, as well as high allelic richness and heterozygosity (Table 2). Travemünde had similar levels of high diversity in terms of polymorphic sites and allelic richness but had only a few private alleles. The lowest diversity was observed in the Danube, closely followed by Mariehamn, both of which had more than 50% fixed alleles. Of the freshwater samples, Elbe showed the highest level of diversity and had the second highest amount of private alleles of all samples. All sample locations, with the exception of Kindvig which had a very small sample size, showed a small degree of heterozygote deficiency. However, none of the samples deviated significantly from expected HW proportions.

No correlations between genetic diversity and shipping density (Pearson correlation: Ar: R = −0.051, *p* = 0.9; P(%): R = −0.34, *p* = 0.4; Pa: R = 0.1, *p* = 0.81; Ho: R = 0.2, *p* = 0.64) or distance to the point of highest shipping density (Pearson correlation Ar: R = 0.29, *p* = 0.49; P(%): R = 0.27, *p* = 0.52; Pa: R = −0.18, *p* = 0.67; Ho: R = 0.53, *p* = 0.17) were found.

Pairwise F_ST_ estimates revealed an overall high and significant genetic divergence between the majority of sampled populations and clearly separated freshwater samples from brackish samples (Figure 4, Table 3). Kindvig was found to not be significantly different from many of the other brackish samples, which could be due to low sample size and lack of power. The only other pairwise comparison which was not significant was between Karrebaeksminde and Guldborgsund. Overall, differentiation among brackish samples (F_ST_ = 0.001–0.18) was lower than among freshwater samples. The Elbe showed high levels of divergence against both the Rhine (F_ST_ = 0.23) and the Danube (F_ST_ = 0.27), whilst the latter two diverged little from each other (F_ST_ = 0.08). The river Elbe was also more similar to brackish samples (F_ST_ = 0.14–0.23) than the river Danube and the river Rhine were (F_ST_ = 0.22–0.38). Of all the brackish samples, Sozopol was the most similar to the freshwater samples (F_ST_ = 0.14–0.25). There was a positive correlation between F_ST_ and salinity (Mantel test, r = 0.33, *p* = 0.006).

In concordance with pairwise F_ST_ measurements, individual-based clustering using sNMF differentiated the brackish samples (blue) from the freshwater (yellow) samples (K = 2 in Figure 5). K = 2 also suggests that the Elbe could be a result of admixture between freshwater and brackish water populations, or be a source to both. Additionally, Sozopol showed signs of similarity to freshwater samples at K = 2. At higher levels of K clusters the Elbe was also the first sample to separate from the rest (see Appendix A). With the exception of Kindvig, Karrebaeksminde and Guldborgsund, adding additional K clusters separated all the geographic samples clearly. Travemünde grouped closer with northern Baltic samples from Turku and Raahe than with the other southern Baltic sites.

The best-supported number of K clusters was K = 9 and was evaluated by selecting the lowest cross-entropy criterion.

To estimate and visualize genetic differentiation among individuals without prior assumptions about the population model, we conducted a PCA (Figure 6). The first two principal components separated the data into five distinct clusters, which corresponded to the clusters identified by sNMF at K = 5 and explained more than 17% of the total variation. The first principal component (PC1, *x*-axis) mainly separated the Rhine and the Danube from the remaining samples, with Sozopol and the Elbe clustering between the other brackish and freshwater samples. The second principal component (PC2, *y*-axis) separated the river Elbe and Mariehamn from each other with the remaining samples in the middle.

## 4. Discussion

### 4.1. Summary

Given that the round goby is invasive in a broad range of salinities [33,67], it is imperative to understand how the ancestral origin of the species contributes to their ability to reproduce within their introduced range. While invasion routes are difficult to uncover, our study shows that reproductive traits of introduced round gobies can be tightly linked to the ancestral, locally adapted ecotypes and the population genomics of the species. Similar to the previous study published by Green, Niemax et al. in 2021 [38], our results show that round gobies with a freshwater ancestry had comparatively low sperm performance in brackish or marine conditions, which may limit these genotypes to freshwater environments. Gobies with a brackish genetic ancestry had the best sperm performance in brackish conditions and are therefore likely to be better able to reproduce in these salinities. These are also the fish most likely to be able to reproduce in the higher salinities that the species is currently expanding into [34].

### 4.2. Separate Ancestral Sperm Ecotypes Fuel the Round Goby Invasion

By analysing the traits of sperm sampled from fish from a freshwater and a brackish water ancestral site, we revealed strong phenotypic signs of local adaptation to the different ancestral salinity conditions. More specifically, for both sites, sperm velocity and motility were superior in their local salinity. Our population genetic analysis also showed strong genetic differences between fish from the ancestral river Danube and the ancestral Black Sea site of Sozopol (Figure 3, Figure 4 and Figure 5a). Though we cannot exclude that there are populations at other ancestral sites that contribute with genetic diversity to the introduced ranges [41], we interpret the sperm trait responses of the fish from Danube and Sozopol (Figure 1) as the ancestral phenotypes from which the introduced populations descend.

Trait responses similar to those the ancestral phenotypes display are clearly visible in the introduced ranges (Figure 2). For example, when sperm were tested in brackish conditions, sperm from fish living in high local salinity (average 15–16 PSU; Sozopol and the southern Baltic sites), showed overall similarly high velocity (site median 101.4–127.6 µm s^−1^). Sperm from fish at all three freshwater sites showed significantly lower velocity in brackish conditions compared to fish from the ancestral Black Sea. In brackish salinity, sperm motility was also significantly lower in fish from the ancestral river Danube and the introduced freshwater site of the Rhine. There was also clear genetic structuring, with fish from introduced freshwater sites being more closely related to fish from the river Danube, while fish in the brackish Baltic Sea were more closely related to fish from the brackish Black Sea (Figure 5a). The structuring was also supported by the Mantel test which showed a clear correlation between genetic distance and salinity. These results show that round gobies from the freshwater and brackish sites are adapted to different environmental salinity conditions.

The fish from the ancestral sites showed strong contrasts in genetic diversity (Table 2), with fish from Sozopol in the Black Sea showing comparatively high allelic richness, a large percentage of polymorphic sites and a high number of private alleles, characteristic of a large and old population. In contrast, and as observed in previous studies [41], individuals from the freshwater river Danube showed an overall low genetic diversity. Brown & Stepien suggested that low genetic diversity might be a result of a repeated founder effect as the species gradually progressed northward along the Danube River, as previously seen in the (also invasive) racer goby, *Neogobius gymnotrachelus* [68]. However, we observed a low genetic diversity in fish from the southern end of the Danube river, suggesting that low genetic diversity is prevalent along the full length of the river. Our data cannot explain why the genetic diversity is so much lower in the river Danube compared to many of the more recently colonised rivers. Possible causes could be low survivability or strong selective forces associated with extreme events, and/or intense fishing pressure in the region [43]. The high diversity of the other freshwater rivers could also be a result of multiple sources and introductions from unsampled populations. Despite the difference in genetic diversity, our results clearly show that freshwater ecotypes are more closely related to each other than to brackish ecotypes, and that they share similar phenotypic responses in their sperm when tested across the sampled salinities.

### 4.3. Individual Sites Provide Details of Eco-Evolutionary Dynamics and Ongoing Local Adaptation

Our aim was to use sperm velocity in the local salinity (Figure 2a, Panel 4) as a proxy for the fit of each population’s sperm traits to their local environment. After analysis and comparison of all different conditions, it is clear that sperm velocity is lower in freshwater conditions, even for locally adapted freshwater fish (the Danube as reference). As such, differences between fish from the ancestral Danube and introduced freshwater sites are better explained by the model comparing sperm velocity in freshwater conditions (Figure 2a, Panel 1), rather than comparing it to the average mean across all sampled sites (Figure 2a, Panel 4). In Panel 1, fish from the Elbe showed a decrease in sperm velocity, pointing to this population being less adapted to the local freshwater conditions, compared to fish from other freshwater sites. This site also shows the strongest genetic similarity to the brackish ecotypes (Figure 4 and Figure 5a). However, this site also has a lot of private alleles, which means that it is likely that the Elbe has an unsampled source population. Fish sampled from the Elbe were sampled close to the busy port of Hamburg. For this site, intense shipping that can carry new genotypes possibly increases the likelihood of outbreeding, with a depletion of ancestrally adapted freshwater genes as a result. Fish from the Elbe also had higher sperm performance than fish from the Rhine and the ancestral Danube in brackish conditions. Interestingly enough, this freshwater site is likely to have connectivity to brackish salinity conditions less than 30 km away. However, it is unknown whether or not round gobies occur along the Elbe towards the coastal conditions where brackish salinities are common, and what these fish’s sperm responses to salinity are. Still, the gradual increase in sperm performance in freshwater with genetic similarity to the ancestral site is further evidence that sperm velocity (and to a degree, sperm motility) in specific salinities have a strong genetic component in the round goby.

Interestingly, our results also provide support that local adaptation continues to occur in the introduced populations in the northern Baltic Sea. Rapid adaptation was suggested by Green et al., 2020 in one of these populations [39], but our comparison in the current paper across more sites shows further evidence. For example, when sperm velocity measured in freshwater is compared to fish from the ancestral river Danube (Figure 2a, Panel 1), fish from brackish sites in the southern Baltic Sea showed a decrease in velocity, while fish from sites in the northern Baltic (that experience salinity conditions of 5 PSU or less) generally showed no difference. Secondly, fish from these northern Baltic sites did not show stronger genetic relationships to each other than to other brackish sites (Table 3), which indicates that adaptation is occurring independently and without gene-flow from freshwater environments.

A third example is also provided by the northern Baltic Sea sites characterized by comparatively low salinity (~5 PSU). Among all introduced brackish sites, Raahe and Mariehamn showed the lowest sperm velocity. These two sites were estimated to be colonized more recently and fish from there appeared less well-adapted than fish from Turku, the third site in the northern Baltic Sea, when that study was conducted in 2015 [39]. Our current study shows that individuals from Mariehamn had comparatively low allelic richness and few polymorphic sites, suggesting a recent bottleneck or a founder event by a small number of individuals. The proposed lack of adaptation at this site [39] could be further impeded due to limited genetic variation [69]. Comparing sperm traits of these populations today could show if sperm velocity has improved over time and would offer another line of evidence of rapid local adaption in reproductive traits [70].

In a fourth example, genomic admixture effects are associated with high sperm velocity. In the southern Baltic site of Travemünde, fish showed the highest sperm velocity in local conditions (brackish, ~16 PSU) (Figure 2, Panel 2 and 4). This site also showed high allelic richness, the second highest number of polymorphic loci (the ancestral Black Sea being higher), as well as high heterozygosity. All of these are signs of strong admixture effects [71]. High genetic diversity increases the chances of having the optimal alleles to match an environment and is therefore associated with establishment success and spread of introduced populations [71,72]. Travemünde is a big shipping port, and our results show this site is likely to have been established from multiple sources (including other populations not sampled for this study) [73]. Though we did not find a correlation between genetic diversity and shipping intensity across our sites, the risk that this diverse and successful population could spread and contribute to more introductions and offer a source of genetic diversity elsewhere is still possible.

### 4.4. Sperm Motility, Relaxed Selection and Plasticity

In several fish species, sperm velocity is the predominant factor affecting fertilization success (and therefore under strong selection) [14,15,74,75,76]. In our data of the introduced populations, sperm velocity matches the local salinity better than sperm motility, which is less tightly linked to the salinity environment and possibly under relaxed environmental selection. Though the two traits are often correlated for a given male, sperm motility is also known to be affected by many different factors, not least of all a male’s energy reserves [77,78,79,80] and readiness to spawn [77,81]. This can potentially confound clear trends between sites, since local conditions such as food availability [82,83], nest availability [84] and population density [85] may influence the number and maturity of a male’s sperm. For example, several brackish sites showed higher sperm motility compared to the ancestral Black Sea site when we compared across brackish test salinities. Since introduced invasive species can experience relaxed competition [86,87], decreased predation [87,88] and little resource limitation [86,87], it is possible that more energy resources are available for investment into sperm in these areas compared to native sites. Similarly, overall low sperm motility in the local salinity of both freshwater and brackish ancestral sites (Figure 2b, Panel 4) could also be a reflection of these fish living under more limiting ecological conditions. Introduced round gobies typically experience rapid population growth and can invest heavily in reproduction [89]. The Baltic Sea is also relatively species poor compared to the Black Sea, where fish species diversity is over 6 times higher (27 species in the Baltic versus 180 in the Black Sea; [90,91]) and ecological competition is likely more prevalent.

Round gobies can reproduce under sperm competition [89], and sneaker morph males have been found in both freshwater [42] and brackish environments [38]. Due to the limits of our fishing methods (cage mesh size: 20 mm, and hook size: 10–14 mm), we are unlikely to have caught any of the potential minute sneaker males previously reported by Marentette et al. [42]. However, we found males with high GSI and light colour in the brackish Black Sea, while these were absent from our sample in the freshwater river Danube. It has been argued that a benign fertilization medium (salt water) allows for increased sperm performance and in turn “easier” access to sneaking opportunities and more pronounced sperm competition in brackish environments [38]. If brackish water allows for increased sperm competition, relaxed selection on female mimicry would be expected. This line of argumentation is supported by the trend in our results, but we advise a solid sampling effort to quantify sneaker male presence (across all male sizes) in different environments before this life-history question can be solved.

For many other euryhaline fishes such as *Tilapia* [25], sticklebacks [24] and killifish [26], which are able to reproduce across wide salinity ranges, sperm traits are often plastic and can be acclimated to the spawning salinity. This is in contrast to gobies, which seem to lack this ability. An experiment that studied short term plasticity in round goby sperm traits did not find any support that gobies of freshwater or brackish origin could acclimate their sperm to swim comparatively faster in novel salinities [38]. A similar experiment of the distantly related sand goby reared in different salinities [16] showed the same results. Together, this suggests that gobies as a clade possibly lack an ability to acclimate their sperm performance to novel salinities, despite their euryhaline ecology. The lack of plasticity is also consistent with the above evidence showing signs of local adaptation and previous studies on trait change in round gobies in the Baltic Sea [39].

### 4.5. Implications for Management

Our sampled genotypes show a pattern of sorting based on environmental differences. This clear environment by genotype match is probably an effect of strong environmental selection, based on reproductive or other physiologically linked traits. Gobies from the introduced rivers show limits to their reproductive output in brackish salinities [38]. Low sperm performance in brackish conditions therefore seem to limit freshwater fish from colonizing brackish environments, at least in the short-term perspective afforded by the current invasion which started in the 1990s. Similarly, though fish from the Baltic Sea can spawn in freshwater, their reproductive output is significantly lower [38]. Likewise, Baltic Sea round gobies have higher expression of the heat shock protein gene hsp70 when acclimated to freshwater as compared to brackish and oceanic salinities, suggestive of higher stress levels in freshwater [92]. Green, Niemax et al., 2021 [38], furthermore showed that mortality for freshwater and brackish fish is higher when acclimated to their opposite salinities. High mortality in juveniles has also been shown in experiments mimicking a ballast discharge scenario into a novel salinity [93]. These lines of evidence strongly suggest that spread of different genotypes is limited by salinity.

Two of our sites in the southern Baltic Sea (Travemünde and Guldborgsund) show the highest observed heterozygosity to the other introduced sites, which could be signs of multiple introductions. In our analysis, we did not find a correlation between shipping intensity and genetic diversity. However, shipping intensity is associated with the appearance and spread of round goby [73] and both sites have shipping harbours within 10 km of the site of catch. Due to the high shipping intensity in the Baltic Sea, it is likely that many round goby populations are genetically connected with complex patterns of connectivity affected by local oceanographic processes [90], ecological processes [90], evolutionary processes [35,39] and anthropogenic activity [73]. Increased sampling efforts across harbors and areas where round gobies have spread naturally could potentially reveal the effects of shipping on genetic patterns in the species, but this requires a concerted effort. As discussed previously, high genetic diversity theoretically increases the probability of an environmental match, which can result in more rapid population growth [69,94] and increased strength of ecological effects [95]. Management and conservation efforts are increasingly taking genetic diversity into account [96,97] particularly in the unique and stressed Baltic Sea [98]. We argue that there are conservation benefits to gain from applying population and seascape genomics to the round goby invasion as well. For example, assessing the risk of spread to a novel site could be done by a detailed sampling of the ecotypes in harbours connected to this site by shipping, and allocating efforts to ballast control where it is most needed.

## 5. Conclusions

Our study shows an example where sperm traits influence contemporary evolutionary patterns, ecology and biogeographic processes in a wild animal of human interest. The invasive round goby has sperm that are phenotypically adapted to either freshwater or brackish salinity conditions. Their sperm traits are tightly linked to the population genomics of the species. Round gobies are separated across their invasive range based on the environmental match of their sperm and are in some regions further locally adapting to prevailing salinity conditions. When assessing introduction risks, the genomic background of the round goby can provide valuable information of which population has the ability to reproduce in a given environment.

## Figures and Tables

**Figure 1 cells-10-01743-f001:**
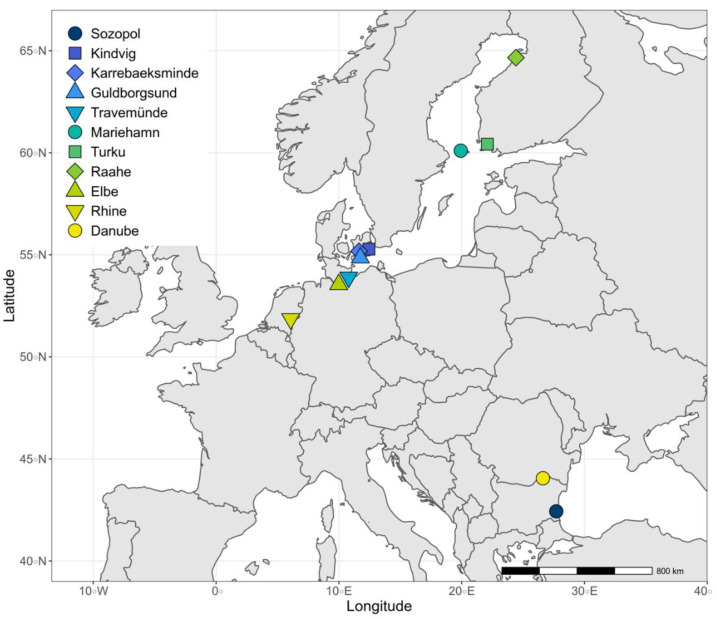
Sites in northern Europe and the ancestral Black Sea region sampled for the invasive fish *Neogobius melanostomus* (see Appendix A for coordinates and associated environmental data). Sozopol (round, dark blue) in the brackish Black Sea, and the river Danube (round, yellow) are ancestral sites sampled for sperm measurements. All other sites represent introduced areas that were sampled during previous studies.

**Figure 2 cells-10-01743-f002:**
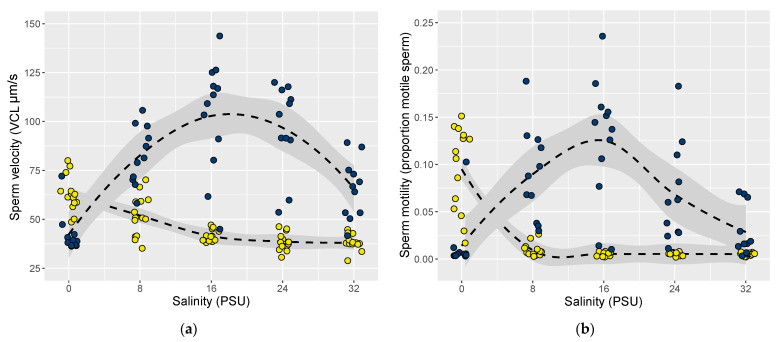
Sperm traits of the invasive fish *Neogobius melanostomus* originating from the ancestral Black Sea site of Sozopol (dark blue) and the ancestral freshwater river Danube (yellow) (see Appendix A for coordinates). Panel (**a**) shows sperm velocity and panel (**b**) shows sperm motility across a salinity gradient of freshwater (0 PSU) to North Sea coastal salinity conditions (32 PSU). Individual data points have been jittered horizontally around each test salinity. Lines show locally weighed scatterplot smoothings (loess smooths) using weighted least squares of the model (response ~ origin x salinity), shaded area shows 95% confidence intervals, both fitted with the ggplot2 package in R.

**Figure 3 cells-10-01743-f003:**
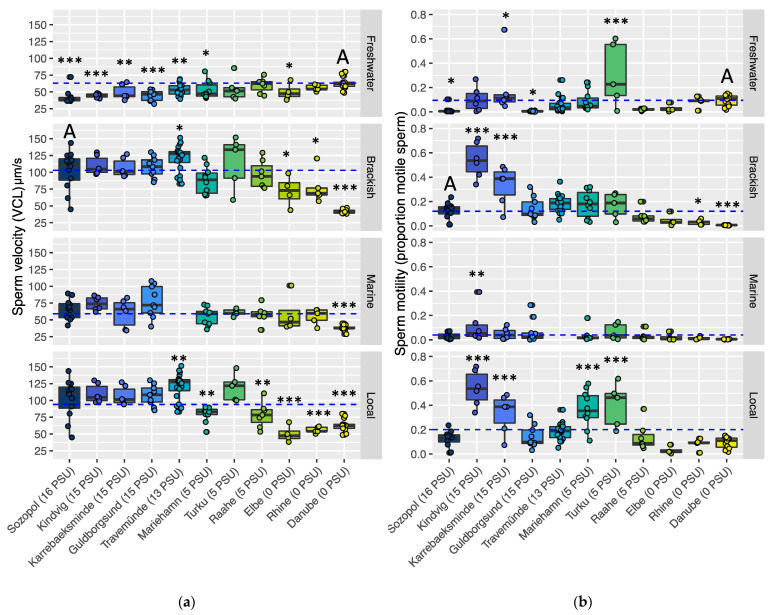
Sperm traits of the invasive round goby (*Neogobius melanostomus*) sampled from the ancestral range (Sozopol, far left, and the Danube, far right on x-axes) and compared against previously published data (all other sites on x-axes) [35,38,39] and across different salinities (facets). Panel (**a**) shows sperm velocity and panel (**b**) shows sperm motility. Facets show different sperm treatment salinities corresponding to freshwater (0–1 PSU), brackish (15–16 PSU), marine (30–32 PSU) and local with varying salinity for each site (see Table 1 for local salinity measurements and estimates). Asterisks indicate significant differences between sites and a mean value (indicated by the dashed line) using a linear model (see Appendix A for full statistics). Means for model tests are: freshwater: the Danube (ancestral freshwater site); brackish: Sozopol (ancestral brackish site); marine: global mean across all sites; local: global mean across all sites. “A” denotes sites of ancestral origin in each respective salinity condition. * = *p* < 0.05, ** = *p* < 0.01, *** = *p* < 0.001.

**Figure 4 cells-10-01743-f004:**
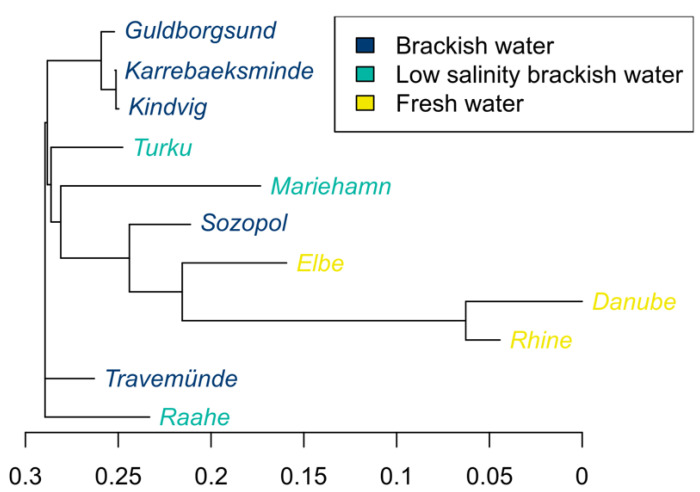
Neighbor-joining tree based on pairwise F_ST_ calculated using 13,847 SNPs showing the phylogenetic relationships between round gobies (*Neogobius melanostomus*) collected from 11 sites. Colour represents environmental condition of population samples.

**Figure 5 cells-10-01743-f005:**
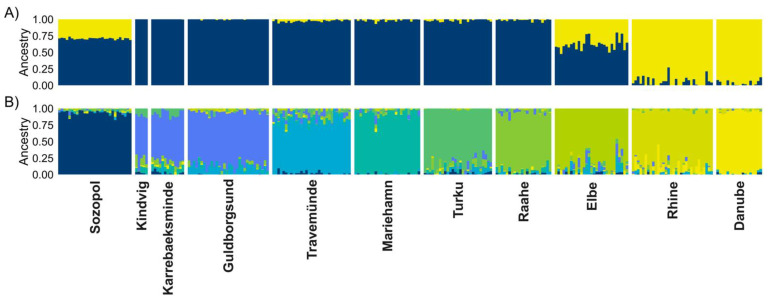
Individual ancestry estimates using sNMF of 264 round gobies (*Neogobius melanostomus*) based on 13,847 SNPs, for (**A**) K = 2 and (**B**) K = 9. Each vertical bar represents one individual; the colour represents the proportion of that individual assigned to the different K clusters. Clusters 3–8 can be found in the Appendix A.

**Figure 6 cells-10-01743-f006:**
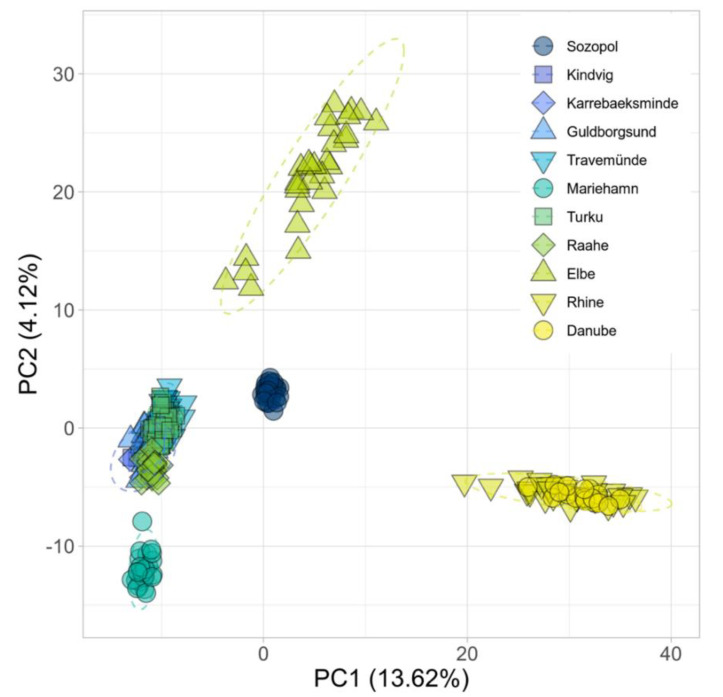
First (*x*-axis) and second (*y*-axis) component of a principal component analysis (PCA) on the genetic relationship of 264 round gobies (*Neogobius melanostomus*) calculated from 13,847 SNPs. The first component explains 13.62% of the total variation and the second explains 4.122%. Each point represents one individual, colours represent sampling origin and shape is used for better distinction between sites. Dashed ellipses show 95% confidence intervals for multivariate normal distributions. The complete visualizations of principal components 1–3 can be found in Appendix A.

**Table 1 cells-10-01743-t001:** Sites, their metadata, sources and summary of phenotypes and genotypes of round gobies (*Neogobius melanostomus*) used for this study. The sites are sorted according to salinity. Coordinates and details of genotype acquisition are available in Appendix A.

Site	Region	Local Salinity (PSU)	Salinities Where Sperm Were Tested (PSU)	Genotypes Kept after Filtering	Data from
Sozopol	ancestral Black Sea	~16	0; 8; 16; 24; 32	N = 29	This study
Kindvig	Southern Baltic	~15	1; 5; 10; 15; 20; 25; 30	N = 5	[39] Green, Havenhand et al., 2020
Karrebaeksminde	Southern Baltic	~15	1; 5; 10; 15; 20; 25; 30	N = 13	[39] Green, Havenhand et al., 2020
Guldborgsund	Southern Baltic	~15	0; 16; 32	N = 32	[38] Green, Niemax et al., 2021
Travemünde	Southern Baltic	~13	0; 16	N = 31	[35] Green, Niemax et al., 2020
Mariehamn	Northern Baltic	~5	1; 5; 10; 15; 20; 25; 30	N = 26	[39] Green, Havenhand et al., 2020
Turku	Northern Baltic	~5	1; 5; 10; 15; 20; 25; 30	N = 27	[39] Green, Havenhand et al., 2020
Raahe	Northern Baltic	~2	1; 5; 10; 15; 20; 25; 30	N = 22	[39] Green, Havenhand et al., 2020
Elbe	western European river	0	0; 16; 32	N = 29	[38] Green, Niemax et al., 2021
Rhine	western European river	0	0; 16; 32	N = 32	[38] Green, Niemax et al., 2021
Danube	ancestral river	0	0; 8; 16; 24; 32	N = 18	This study

**Table 2 cells-10-01743-t002:** Summary statistics of genetic diversity in the round goby (*Neogobius melanostomus*) sampled from two ancestral (Sozopol and the Danube) and nine introduced sites, using 13,847 SNPs.

Site	N	Ar	P(%)	Pa	H_o_	H_e_	F_IS_
Sozopol	29	1.6	80	521	0.19	0.21	0.077
Kindvig	5	1.5	50	0	0.19	0.20	−0.065
Karrebaeksminde	13	1.5	62	0	0.18	0.20	0.021
Guldborgsund	32	1.5	68	2	0.20	0.20	0.019
Travemünde	31	1.6	74	7	0.20	0.21	0.020
Mariehamn	26	1.4	49	1	0.16	0.17	0.038
Turku	27	1.5	68	9	0.19	0.20	0.031
Raahe	22	1.5	62	0	0.17	0.19	0.041
Elbe	29	1.5	68	182	0.19	0.20	0.018
Rhine	32	1.4	66	56	0.16	0.16	0.015
Danube	18	1.3	44	4	0.13	0.13	0.018

N = sample size, Ar = allelic richness, P(%) = percent polymorphic sites, Pa = private alleles, H_o_ = observed heterozygosity, H_e_ = unbiased expected heterozygosity, F_IS_ = inbreeding coefficient.

**Table 3 cells-10-01743-t003:** Pairwise F_ST_ of 11 population samples of round goby (*Neogobius melanostomus*) using 13,847 SNPs, calculated according to Weir and Cockerham (1984). Bold values indicate significance of the estimates and were tested using Fisher’s exact tests with 5000 Monte Carlo replicates and false discovery rate corrections.

	Sozopol	Kindvig	Karrebaeks- Minde	Guldborg-Sund	Trave- Münde	Mariehamn	Turku	Raahe	Elbe	Rhine
**Kindvig**	**0.113**									
**Karrebaeksminde**	**0.115**	0.001								
**Guldborgsund**	**0.121**	0.019	0.012							
**Travemünde**	**0.106**	0.068	**0.070**	**0.062**						
**Mariehamn**	**0.179**	0.154	**0.147**	**0.147**	**0.144**					
**Turku**	**0.113**	**0.072**	**0.074**	**0.082**	**0.070**	**0.154**				
**Raahe**	**0.138**	0.092	**0.093**	**0.095**	**0.083**	**0.173**	**0.097**			
**Elbe**	**0.139**	0.169	**0.168**	**0.158**	**0.142**	**0.232**	**0.160**	**0.194**		
**Rhine**	**0.218**	**0.308**	**0.294**	**0.275**	**0.261**	**0.339**	**0.282**	**0.304**	**0.230**	
**Danube**	**0.249**	**0.369**	**0.342**	**0.320**	**0.307**	**0.383**	**0.321**	**0.345**	**0.269**	**0.081**

## Data Availability

The phenotypic data presented in this study are openly available in DataDryad at doi:10.5061/dryad.6djh9w11q.

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
