# Peer review of "Ancestral Sperm Ecotypes Reveal Multiple Invasions of a Non-Native Fish in Northern Europe"

_cells, 2021, doi:10.3390/cells10071743_

Round 1

Reviewer 1 Report

Please see word attachment of general interest to the journal's readership.

Reviewer 2 Report

This is an interesting study that links sperm performance and genotypic diversity in relation to salinity of ancestral brackish and freshwater populations and other populations of the externally fertilizing invasive round goby. The study is sound and the manuscript well written. I have some suggestions for improvement.

- the title is not very catchy and could be improved

- write species and genus names in italic throughout the text

Introduction:

I have missed predictions for your experiments. For sperm performance, there are clear predictions based on similar experiments with other populations in Green et al. 2019, 2020, 2011. Due to the descriptive nature, it is less common to make predictions for genetic diversity, but still possible in your case. One factor that influences admixture is shipping activity as you mention in the Discussion. Provide data on shipping activity at or in close surroundings of the various sampled locations. The authors should clearly mention what their study distinguishes from the studies of Green et al. (2019, 2020, 2021) and more clearly discuss the similarities with the results of the Green et al. (2019, 2020, 2021) studies.

Methods:

- P4, L159: give volume of the salinity solution

- give male body size of the different populations in Table 1 and give the number of sperm in 25 μl samples. Sperm density may have influences on sperm velocity and sperm motility

- why were so many genotypes disregarded after filtering in some populations (Kindvig, Danube) (Suppl. Table 1)? Are the used genotypes in these populations still representative?

- Have differences in test salinities between population systematically contributed to differences in performance between populations?

- A map of the sampled locations would be very helpful

Results:

P7, L311: change "may be" into "is"

P7, L320: delete "Estimates of sperm velocity are presented in Figure 1c." (there is no Fig. 1c)

P8, L326: delete "Estimates of sperm motility are presented in Figure 1d." (there is no Fig. 1d)

P8, L331: "points"

P8, L332: what is "loess smooths". Explain in the Methods

- Fig. 1: why is sperm motility so low? Are the numbers comparable to other studies in gobies and other fishes?

P9, L358: renumber Fig 1 into Fig. 2, and check Fig. numbers throughout the text

- Fig. 1 (=2): Populations should be arranged to declining PSU (Guldborgsund should be placed before Kindvig), or is there a reason why this was not strictly done. This also concerns all other Figs and Tables

- Fig. 1 (=2): for what stands "A" in the panels (4 times)

- Fig. 1 (=2): write μ on the y-axis in normal font (no italic) as in other places in the text

P11, L439 and P12, L461: add a period after "Fig"

- Table 2 and Table 3: write decimal points instead of commas

P12: renumber Figure 2 into Figure 3

- Fig. 2 (=3): add Fig. legend and cite Fig. 3 in the text

- legend of Table 3: write "MC reps" in full

- Table 3: how do you interpret that the pairwise Fst-values correlate strongly with salinity? Please analyse

- legend of Fig. 5: mention the meaning of the broken lines, and indicate that the PCA was on genetic differentiation

- Fig. 5: label the axes PC1 and PC2 instead of giving percentages (also in Suppl. Materials)

- Suppl. Table 1: write "Coordinates"

- Suppl. Information: place the Fig. legends below the Figs instead of above
